# Improving Sesame (*Sesamum indicum* L.) Seed Yield through Selection under Infection of *Fusarium oxysporum* f. sp. *sesami*

**DOI:** 10.3390/plants11121538

**Published:** 2022-06-09

**Authors:** Rasha Ezzat Mahdy, Dalia A. Gaber, Mohamed Hashem, Saad Alamri, Ezzat E. Mahdy

**Affiliations:** 1Plant Breeding, Agronomy Department, Faculty of Agriculture, Assiut University, Asyut 71515, Egypt; emahdy@aun.edu.eg; 2Botany and Microbiology Department, Faculty of Science, Assiut University, Asyut 71515, Egypt; dalia_gaber@aun.edu.eg; 3Department of Biology, College of Science, King Khalid University, Abha 61413, Saudi Arabia; mhashem@kku.edu.sa (M.H.); saralomari@kku.edu.sa (S.A.)

**Keywords:** pedigree selection, restricted selection, observed genetic gain, heritability

## Abstract

Sesame (*Sesamum indicum* L.), the Queen of oilseeds, is infected with different pathogens, restricting its yield. *Fusarium oxysporum* f. sp. *sesami* is the most destructive disease of sesame worldwide, causing economic losses. This work aimed to develop new high-yielding strains, resistant and/or tolerant to *Fusarium*. Two cycles of pedigree selection were achieved under infection of *Fusarium oxysporum* f. sp. *sesami*. Two populations in the F_2_ (600 plants each) were used. The selection criteria were five single traits and another three restricted by yield. The restricted selection was better in preserving variability than the single trait selection. The observed genetic gain in percentage from the mid-parent in the F_4_-generation was significant for the eight selection criteria. Single trait selection proved to be an effective method for improving the selection criterion, but it caused deleterious effects on the other correlated traits in most cases. The seed yield increased by 30.67% and 20.31% from the better parent in the first and second populations, respectively. The infection% was significantly reduced by 24.04% in the first, and 9.3% in the second, population. The selection index improved seed yield, and its attributes can be recommended.

## 1. Introduction

Sesame (*Sesamum indicum* L.) is an annual plant of the *Pedaliaceae* family and one of the most ancient oilseed crops known to man. Its cultivation goes back to 2130 B.C. [1]. However, it was suggested that it goes back to 3050–3500 B.C. [2]. It is adapted to grow in tropical and sub-tropical areas. Sesame has been grown in the Near East and Africa for over 5000 years, for cooking and medicinal needs [3]. Generally, 65% of world sesame production is used as edible oil, and 35% for confectionary purpose. The fatty acid composition contains high levels of unsaturated fatty acids, and low levels of saturated (less than 15%) and antioxidant fatty acids [4,5,6]. The presence of antioxidants has been reported to have health promoting effects, such as lowering cholesterol levels and hypertension in humans and to have neuroprotective effects against hypoxia and brain damage [7], as well as reducing the incidence of certain cancers [8,9]. Sesame seed is a source of protein and high in sulfur, containing amino acids, minerals, and vitamins [10,11]. Sesame oil compounds also have multiple physiological functions, such as estrogenic activity, providing anti-inflammatory functions, and decreasing blood lipids and arachidonic acid levels [12].

Sesame is attacked by at least eight economically important fungal diseases, from the seedling stage to maturity, but the most drastic disease is sesame wilt, caused by *F. oxysporum* f.sp. *sesami*, which limits the production of sesame and causes yield losses of up to 100% [1,2]. It is a soil-borne root pathogen blocking xylem vessels, resulting in wilt. It was first reported in the USA in 1950 [13]. Histopathological studies [14] indicated that the pathogen moved intercellularly adjacent cortical and epidermal cells, causing disintegration of the cells and blocking xylem vessels with gum, which retards water and nutrient supply to plants; thus, resulting in wilting and death of the plant.

The area of sesame cultivation in Egypt has tended to decrease due to disease infection. Sesame is a low requirement crop and responds well to additional inputs of water and nutrition [15]. To overcome this problem, developing resistant genotypes by breeding is a must. Evaluation and selection for resistance to *Fusarium* in a contaminated field could improve resistance. Significant differences in susceptibility and resistance to *Fusarium* among 25 genotypes over two years were found by [16]. While, [17] evaluated 35 sesame genotypes and observed that all the accessions showed different infection rates, and none of them were immune. In addition, 41 sesame genotypes were evaluated under severe infection of *Fusarium*, and none could be described as immune [16,18]. Four displayed a higher rate of germination (14–16%), five expressed a lower rate of germination (less than 2%), and seven could not germinate in the contaminated soil. Additive and non-additive gene actions, and reciprocal effects, were involved in the inheritance of resistance to *Fusarium* wilt, and selection for resistance to *Fusarium* should be done in later generations [19]. Selection in late generation was better for improving yield than in the early generations; one cycle of selection for seed yield in sesame started in the F_4_- generation exceeded two cycles in the F_2_ and F_3_ -generations, by 10.38% and 24.51%, in two populations [20]. Two cycles of pedigree selection started in the F_2_ decreased genotypic and phenotypic variations, significantly lowered height to first capsule, and increased the observed genetic gains of plant height, as well as the number of capsules plant^−1^ (43.21% and 81.52%) and seed yield in two populations (30.68% and 45.18% of the better parent) [21]. Number of capsules plant^−1^ and 1000 seed weight could be used as selection criteria to increase seed yield [17,22,23,24]. This work aimed to improve seed yield, tolerance, and/or resistance to *F. oxysporum* through selection of eight criteria; earliness, plant height, height to the first capsule, length of fruiting zone, and seed yield; with earliness, plant height, and length of fruiting zone restricted by seed yield, under artificial infection of *F. oxysporum*.

## 2. Result

### 2.1. Description of the Base Population (F_2_ Generation)

Mean days to 50% flowering in the F_2_- generation of pop1 was later than the respective parents. However, it was earlier than the earlier parent in pop2 (Table 1). Means of PH (plant height) and LFZ (length of the fruiting zone) in the base populations were better than the parents. Mean HFC (height to the first capsule) showed partial dominance towards the higher parent in the F_2_- generation of pop1 *(Shandaweel 3/Sohag2000****)***, and towards the lower parent in pop2 (*Int.562/Int.688)*. Mean SYP^−1^ (seed yield plant^−1^) showed no dominance in pop1 and complete dominance towards the lower parent in pop2. The infection percentage in F_2_-generation of pop1 showed partial dominance towards susceptibility, and absence of dominance in the other base population.

The PCV% was 8.42% for earliness, 13.61% for PH, 25.90% for HFC, 20.56% for LFZ, and 59.17% for SYP^−1^, indicating sufficient variability and feasibility of selection in pop1. The other base population also showed sufficient variability. Estimates of heritability in the broad sense varied in the two populations. In pop1 it ranged from 29.89% for days to 50% flowering to 77.72% for HFC, and from 10.32% for HFC to 75.20% for days to 50% flowering in pop2. The expected genetic advance from selection of 10% of superior plants in pop1 varied from 4.43% for flowering to 35.42% of the mean for HFC, and from 5.85% for HFC to 84.50% of the mean for SYP^−1^ in pop2. Estimates of heritability differed greatly in both populations, accompanied by differences in the expected genetic advance from selection.

### 2.2. The Phenotypic Correlations among Traits of the Base Population (F_2_ Generation)

Days to 50% flowering (Table 2) showed negative significant (*p* ≤ 0.01) correlations with PH, LFZ, and SYP^−1^ in pop1, indicating that the late plants were short in height, LFZ and low yielding. Likewise, earliness gave a negative significant correlation with SYP^−1^ in pop2 and positive with HFC. In both populations, 1000 sw negatively correlated with all traits. The main features of the correlations were the positive significant (*p* ≤0.01) correlations among PH, HFC, LFZ, and SYP^−1^ in both populations, indicating that the tall plants were tall in the fruiting zone and had a high yielding ability. Therefore, PH and LFZ should be considered for improving seed yield in these materials.

### 2.3. Variability and Heritability after Two Cycles of Selection (F_4_-Generation)

The entries mean squares (five selected families + two parents) were significant (*p* ≤ 0.01) for all traits in both populations (Table 3).

The genotypic coefficient of variation (Table 4) was depleted after two cycles of pedigree selection for earliness (3.68% for pop1 and 5.70% for pop2); meanwhile, it was high in pop1 (14.97%) and moderate in pop2 (7.09%) when selection was performed for earliness restricted by yield (FLOWR). Generally, the genetic variability tended to decrease from F_2_ to F_4_- generation, especially for LFZ and SYP^−1^. However, the remaining genetic variability in the selection criteria in the F_4_-generation was sufficient for further cycles of selection for HFC, LFZ, SYP-1, and 50% flowering, and PH and LFZR, in both populations. The inclusion of SYP^−1^ as a restricted trait with earliness, PH, and LFZ tended to preserve variability. The heritability in the broad sense for the selection criteria was very high and unreliable, and ranged from 87.92% for plant height in pop2, to 97.80% for SYP^−1^ in pop1 (Table 5). However, the narrow-sense heritability (h^2^), as estimated by parent-offspring regression, of F_4_/F_3_ was low to moderate. It was 0.48 and 0.49 for days to 50% flowering, 0.40 and 0.37 for PH, 0.44 and 0.42 for HFC, 0.23 and 0.22 for LFZ, and 0.29 and 0.22 for SYP^−1^ in pop1 and pop2, respectively.

### 2.4. Direct and Correlated Observed Gain after Two Cycles of Pedigree Selection (F_4_-Generation)

#### 2.4.1. Selection for Days to 50% Flowering (Earliness) and Earliness Restricted by Yield (FLOWR)

The direct response in earliness after two cycles of selection was favorable and significant (*p* ≤ 0.05–*p* ≤ 0.01). It was −4.96 and −2.33% for pop1, and −8.59 and −6.56% for pop2, from the mid and the earlier parent, respectively (Table 6). Selection for earliness improved HFC by −14.74 and −11.76%, and infection % by −25.16 and −8.49% from the mid-parent for pop1 and pop2, respectively. However, selection for earliness significantly (*p* ≤0.01) decreased the correlated genetic gains of NCP^−1^, by −31.85 and−36.36% in pop1, and by −27.96 and −30.91% in pop2 from mid and better parents, respectively. Furthermore, PH was decreased by −6.87 and −11.80% from the taller parent, and SYP^−1^ by −15.64 and −13.88% from the high yielding parent in pop1 and pop2, respectively. Selection for FLOWR (the early families that gave a seed yield plant^−1^ equal to or more than the population mean) did not affect earliness, but it was better at improving PH, LFZ, NCP^−1^, and SYP^−1^ in both populations than selection for earliness itself.

#### 2.4.2. Selection for Plant Height (PH) and PH Restricted by Yield (PHR)

The direct observed genetic gain in PH was significant (*p* ≤ 0.05–*p* ≤ 0.01) and reached 20.31 and 8.3% from the mid-parent, and 16.72 and 9.30% from the better parent in pop1 and pop2, respectively. Single trait selection for PH was efficient in improving some correlated traits; HFC, LFZ, NCP^−1^, SYP^−1^, and infection %. The correlated response was −14.87 and −11.76% for HFC, 40.95 and 24.62 for LFZ, 20.64 and 14.23% for NCP^−1^, 22.13 and 14.23% for SYP^−1^, and −20.77 and −9.17% from the mid-parent for infection% in pop1 and pop2, respectively. However, selection for PH significantly delayed earliness by 16.06% from the mid-parent and 19.26% from the earlier parent in pop1, and 6.09% from the mid-parent and 8.44% of the earlier parent in pop2. The observed genetic gain from selection for PHR was better in improving PH, LFZ, NCP^−1^, and SYP^−1^ than selection for PH per se.

#### 2.4.3. Selection for Height to the First Capsule (HFC)

The direct observed genetic gain in HFC was significant (*p* ≤ 0.05–*p* ≤ 0.01) and reached −10.13 and −13.38% from the mid-parent in pop1 and pop2, respectively (Table 6). A favorable indirect correlated gain in earliness and infection% was obtained. It was −6.30% of the mid-parent for earliness in pop2, and −26.98 and −7.02% for infection % in pop1 and pop2, respectively. However, deleterious effects were observed in the other correlated traits. The significant (*p* ≤ 0.05–*p* ≤ 0.01) decrease from the mid-parent was −13.17 and −11.09% for PH, −37.39 and −11.31% for NCP^−1^, −18.99 and −14.96% for SYP^−1^, and −12.74 and −9.90% for a 1000 seed weight in pop1 and pop2, respectively.

#### 2.4.4. Selection for the Length of Fruiting Zone (LFZ) and LFZ Restricted by Yield (LFZR)

The direct genetic response to selection for LFZ was significant (*p* ≤ 0.05–*p* ≤ 0.01). It was 27.22 and 26.47% in pop1, and 20.77 and 18.17% in pop2 from the mid and better parent, respectively. Selection for LFZ insignificantly lowered HFC. The favorable significant indirect response for NCP^−1^ was 13.73 and 17.11%, 19.51 and 16.28% for SYP^−1^, and −26.61 and−11.58% for infection % from the mid-parent in pop1 and pop2, respectively. The observed genetic gain from selection for LFZR was the best method for improving LFZ, NCP^−1^, and SYP^−1^ in both populations.

#### 2.4.5. Selection for Seed Yield Plant-1 (SYP^−1^)

The direct genetic response to selection for SYP^−1^ (Table 6) was significant (*p* ≤ 0.01) from the mid-parent in both populations. It was 24.72 and 17.43% in pop1 and pop2, respectively, and significant from the better parent (10.07%) in pop2. However, selection for SYP^−1^ was accompanied with an unfavorable significant increase in days to 50% flowering by 4.0 and 3.15% from the mid-parent, and 6.87 and 5.44% from the earlier parent, in both populations. A favorable indirect insignificant response was observed for HFC and LFZ in pop1, and as significant in LFZ in pop2. The significant correlated gain in NCP^−1^ was 14.54 and 15.12% from the mid-parent, and 8.81 and 10.41% from the better parent in pop1 and pop2, respectively. Selection for SYP^−1^ caused a significant (*p* ≤ 0.01) decrease in infection % of −25.16 and −10.04% from the mid-parent in both populations.

## 3. Discussion

### 3.1. Variability and Heritability before and after Selection

Sesame is liable to be attacked by many economically important fungal root rot diseases from the seedling stage to maturity [22]. *F. oxysporum* is a soil-borne root pathogen [13], the most damaging disease, and can cause severe economic losses [2]. As it is a soil-borne disease, once it is noticed in a field it cannot be easily controlled. Unfortunately, very little is known about the existence of reliable sources of resistance. Therefore, selection under artificial infection is an appropriate method to improve yield and tolerance to *Fusarium*. Indeed, selection for yield and its attributes under artificial infection is also selection for resistance to *Fusarium*. In other words, the plants are under the pressure of *Fusarium* and under the pressure of selection criteria.

Selection mainly depends upon the presence of genetic variability in the traits concerned. The variability in the F_2_ of the two base populations (Table 1) showed that the range (minimum and maximum value) of the five-selection criteria (earliness, PH, HFC, LFZ, and SYP^−1^) was wider in the two base populations than in their respective parents, indicating the feasibility of selection, and some individuals showed transgressive segregation. Except for earliness, the PCV in SYP^−1^ was considered high, and reached 59.17 and 72.17% in pop1 and pop2, respectively (Table 1). High estimates of heritability resulted in large estimates of expected genetic advance in the percentage of the mean, except for earliness in pop1. The authors of [20,25,26] noted that a wide variability coupled with high estimates of heritability resulted in high estimates of expected genetic advance.

In the F_4_-generation, after two cycles of selection (Table 4), the genetic variability in the selection criteria decreased. However, the remaining genetic variability in the selection criteria was sufficient for further cycles of selection for HFC, LFZ, and SYP^−1^. Restricting selection criteria by yield (FLOWR, PHR, and LFZR) preserved genetic variability better than single trait selection. Such a decrease is expected, because the selection was for the upper part of the curve, in which the selected plants became more similar. It is well known that selection favors heterozygosity. Reference [27] noted that in self-pollinated crops selection cycle after cycle increased homozygosity; in consequence, the selection differential and genetic variance decreased. These results agree with those reported by [18,21,27,28,29].

The unreliable estimates of heritability in the broad sense (Table 5), in the F_4_-generation for the selection criteria, could be due to the evaluation of the selected families in one site for one year, which inflates families mean squares by the interactions of families × locations, families × years, and families × year × locations [22]. However, the narrow-sense heritability (h^2^), as estimated by parent–offspring regression, of F_4_/F_3_ was low to moderate. The low h^2^ was for SYP^−1^ and LFZ, while it slightly increased for HFC, PH, and 50% flowering. After two cycles of selection, [27,29] noted high estimates of broad-sense heritability for HFC, NCP^−1^, and SYP^−1^. These results agree with those reported by [26], who noted a high broad-sense heritability of 84.65% for PH, 94.89% for HFC, and 96.59% for SYP^−1^.

### 3.2. Correlations among Traits in the Base Populations (F2-Generation)

Correlation among traits is an important tool, required to identify selective traits in a selection program. Pleiotropy and linkage are the main genetic causes of correlation. The negative correlation of days to 50% flowering (Table 2) with PH, LFZ, and SYP^−1^ adversely affected and reduced the observed gains in these traits (Table 6), in both populations. However, the positive significant (*p* ≤ 0.01) correlations among PH, LFZ, and SYP^−1^ in both populations were reflected in the improvements in these materials. These results are in line with those reported by [22,30,31,32].

### 3.3. Direct and Correlated Observed Gain after Two Cycles of Pedigree Selection (F_4_-Generation)

#### 3.3.1. Selection for Days to 50% Flowering (Earliness) and Earliness Restricted by SYP^−1^; FLOWR

The direct observed gain in days to 50% flowering (earliness) after two cycles of selection was significant (*p* ≤ 0.05–*p* ≤ 0.01) from the mid-and the earlier parent (Table 6), proving the efficiency of pedigree selection in improving the selection criterion. Selection for earliness improved HFC, because of their positive correlation. Otherwise, it caused a deleterious decrease in the important correlated traits in sesame; PH, NCP^−1^, and SYP^−1^.

Otherwise, selection for FLOWR did not affect earliness and significantly increased PH (8.92%), LFZ (24.26%), NCP^−1^ (11.11%), and SYP^−1^ (17.53%), and decrease infection% by −26.61% from the mid-parent. Hence, SYP^−1^ as a restricted trait, when selected for earliness, mitigated the effects of the negative correlations of earliness with the other traits, and significantly increased the observed genetic gains in PH, LFZ, and SYP^−1^. As it was found that selection for FLOW(R) was better than selection for earliness per se [33].

#### 3.3.2. Selection for Plant Height (PH) and PH Restricted by SYP^−1^; PHR

Direct observed genetic gain in PH was significant (*p* ≤ 0.05–*p* ≤ 0.01) and reached 20.31 and 8.3% from the mid-parent, and 16.72 and 9.30% from the better parent in pop1 and pop2, respectively. The correlated gains were favorable for LFZ, NCP^−1^, SYP^−1^, and infection %. These results are in accordance with those noted by [21,33]. However, selection for PH significantly delayed earliness. Selection for PHR showed better improvement of LFZ, NCP^−1^, and SYP^−1^ from the mid-parent in both populations. Selection for PH ranked second in improving SYP^−1^, after selection for LFZR. Furthermore, a significant (*p* ≤ 0.01) reduction in infection% from the mid-parent of −22.67 and −10.04% in pop1 and pop2, respectively, was obtained. These results were expected, because of the highly significant correlations among the three traits (PH, LFZ, and SYP^−1^) in the base populations. These results agree with those reported by [20,24,25], who indicated that yield improvement would be possible via selection for number of number of capsules plant^−1^, diameter of stem, capsule length, number of branches per plant, and plant height.

#### 3.3.3. Selection for Height to First Capsule (HFC)

The direct observed genetic gain in HFC was significant (*p* ≤ 0.05–*p* ≤ 0.01) and reached −10.13 and −13.38% from the mid-parent in pop1 and pop2, respectively (Table 6). A favorable indirect correlated gain in earliness and infection% was obtained. However, a deleterious significant (*p* ≤ 0.05–*p* ≤ 0.01) decrease was obtained for PH, NCP^−1^, SYP^−1^, and for 1000 seed weight. This could be due to the positive correlation between HFC, PH, and SYP^−1^. Therefore, selection for low HFC in these materials was the worst method for improving seed yield. In the field, the plants with low HFC were shor; in consequence, short in the fruiting zone with a low number of capsules and low yielding ability. Ref. [19] noted gains in HFC of −56.71% and −56.17% of the better parent for two populations, after two cycles of selection.

#### 3.3.4. Selection for Length of Fruiting Zone (LFZ) and LFZ Restricted by SYP^−1^; LFZR

The positive significant correlations among PH, LFZ, and SYP^−1^ in the base populations played an important role in pedigree selection [21,23,26,33]. The direct genetic response to selection for LFZ was significant (*p* ≤ 0.05–*p* ≤ 0.01). Selection for LFZ showed a favorable significant indirect response from the mid-parent for NCP^−1^ (13.73% **, 17.13% **, SYP^−1^ (19.51% **, 16.28% **), and infection % (−26.61% **, −11.58% **) in pop1 and pop2, respectively. However, selection for LFZ significantly delayed earliness in pop1. Selection for LFZR was the best out of the eight methods for improving LFZ (30.83% **, 24.94% **), SYP^−1^ (34.04% **, 28.53% **) infection %, and HFC (−15.13% *, −12.03%) from the mid-parent in pop1 and pop2, respectively. This could be due to the significant positive correlation among LFZ, PH, and SYP^−1^. Ref. [26] indicated that SYP^−1^ had a significant correlation with LFZ in branched and non-branched genotypes. While, a high SYP^−1^ from selection for LFZ restricted by HFC was noted by [20]. 

#### 3.3.5. Selection for Seed Yield Plant^−1^ (SYP^−1^)

The direct genetic response to selection for SYP^−1^ was significant (*p* ≤ 0.01) from the mid-parent in both populations. A favorable indirect response was observed for HFC, LFZ, NCP^−1^, and infection%. Selection for SYP^−1^ significantly (*p* ≤ 0.01) decreased infection%, by −25.16 and −10.04% from the mid-parent in pop1 and pop2, respectively. However, selection for SYP^−1^ accompanied unfavorable indirect effects in days to 50% flowering in both populations. One cycle of selection for seed yield in sesame, started in the F_4_ generation, exceeded two cycles in the F_2_ and F_3_ generations by 10.38% and 24.51% in two populations [20]. An observed gain was recorded by [21] in SYP^−1^, of 30.68% and 45.18% for the better parent in two populations. It was found that the observed genetic gain in seed yield plant^−1^ after two cycles of selection was significant (*p* ≤ 0.01), and reached 80.27% of the unselected bulk sample [26].

## 4. Methods

### 4.1. Experimental Design and Procedures

Two cycles of pedigree selection were carried out on two segregated populations of sesame (*Sesamum indicun* L.) in the F_2_, F_3_, and F_4_ generations, in the three successive seasons of 2018, 2019, and 2020, at the Faculty of Agriculture Experimental Farm, Assiut University, Egypt (Longitude: 31.125 N, Latitude: 27.25 E, Elevation: 45 m/148 Feet). The segregating populations were *Shandaweel*3/*Sohag2000* (pop1/600 plants) and *Int.562/Int.688* (pop2/600 plants). The parents were *Shandaweel*3 cultivar traced back to ‘*Giza32 × Introduction 130*’ and released in 1992, *Sohag2000* originated from the cross ‘*Toshka1 × Introduction 416*’ and was released in 1992, *Introduction 562* was imported from FAO in 1983, and *Introduction 688* was imported from Israel in 1988. These parents are considered high yielding. The experimental site was inoculated each year by *F. oxysporum* f. sp. sesami (Zap) Cast. The soil was clay (sand 27.4%, silt 24.3%, clay 48.3%), EC (1:1 extract) dSm^−1^ 1.07, pH 8.1, organic matter 0.24%, soluble K (mg kg^−1^) 39, soluble Ca (mg kg^−1^) 190, soluble HCO_3_ (mg kg^−1^) 427, soluble Na (mg kg^−1^) 140, total nitrogen 0.08%, and soluble Mg (mg kg^−1^) 72.

The planting date in the three seasons ranged from April 12th to 15th. In the first year the F_2_ seeds were sown in rows 4 m long, 60 cm apart, and 10 cm within a row (Figure 1). Seeds were dibbled in holes and covered with an equal amount (approximately 40 g) of pathogen inoculum at the ratio of 1% soil weight. The two populations and the parents were sown in 15 rows each. After full emergence, seedlings were thinned to one plant hill^−1^. The recommended culture practices for sesame production were adopted throughout the growing season. The recorded traits on 100 guarded survival plants were days to 50% flowering (earliness), plant height (PH, cm), height to the first capsule (HFC, cm), length of the fruiting zone (LFZ, cm = PH-HFC), number of capsules plant^−1^ (NCP^−1^), seed yield plant^−1^ (SYP^−1^, g), and infection % (Inf % = 1−(number of survival plants at harvest/number of seedlings after thinning) × 100). The traits of the parents were recorded in 30 survival plants. The best 10 plants for each of eight criteria; days to 50% flowering, days to 50% flowering restricted by seed yield plant^−1^ (FLOWR) (the early flowering plants which gave seed yield plant^−1^ equal to or more than the population mean), plant height, plant height restricted by yield (PHR), height to the first capsule, length of fruiting zone, length of fruiting zone restricted by yield (LFZR), and seed yield plant^−1^ were saved from each population for the second season. In the second season, the selected plants of each population and parents were sown in families in a randomized complete block design (RCBD) with three replications. The distances between rows and hills were as in the previous season. The plot size was three rows. The best plant from each of the best five families for each selection criterion was saved as the second cycle selections. In the third season (F_4_), the five selected families for each selection criterion, along with the parents, were evaluated in RCBD, as in the previous season. Oil percentage was measured as outlined by the standard method of A.O.A.C. (1980), but it was deleted from the tables because of the insignificant differences among families.

### 4.2. Isolation and Identification of the Causal Pathogen

*Fusarium oxysporum* f. sp. *sesami* (Zap) Cast. was isolated from sesame plants showing symptoms of wilt. Diseased tissues were cut into small pieces (2–3 mm), washed thoroughly with tap water, the surface was sterilized with 3% NaOCL for 3 min, and washed two times with sterilized water. The sterilized pieces were cultured in Potato Dextrose Agar (PDA) and incubated for 3–5 days.

Identification of the isolated fungi was carried out on 5–12-day-old culture, using the morphological and microscopic characteristics of mycelium and spores according to [12]. During planting, sesame seeds were dibbled in holes and covered with an equal amount (approximately 40 g) of pathogen inoculum at a ratio of 1% soil weight.

### 4.3. Statistical Analysis

The analysis of variance (ANOVA), and phenotypic (σ^2^p) and genotypic variance (σ^2^g) were performed as in [34]. The analysis of variance was performed twice. The first was for all entries (selected families + parents), the second was for the selected families only, to estimate coefficients of variation and heritability (not included), as these should be calculated without the parents. The analysis of variance of the infection percentage was carried out on Arcsine transformed data.
Genotypic variance (σ^2^g) = (Msg − MSe)/r
Phenotypic variance (σ^2^g) = σ^2^g + MSe/r
where, MSg = genotypes mean square, MSe = error mean square, r = number of replications.

The phenotypic and genotypic coefficients of variation were estimated using the formula developed by [35], which is used to identify the usefulness of the variation, as being sufficient for reselection or not.
GCV% = (σg/mean) × 100; and PCV% = (σp/mean) × 100
where σg and σp = genotypic and phenotypic standard deviations, respectively.

Heritability in the broad sense (H) and the genetic advance were computed using the formula adopted by [27], as follows:Heritability in the broad sense (H%) = (σ^2^g/σ^2^_p_) ×100 and the expected genetic gain in the F_2_ = k* σp*
where the environmental variance σ^2^_E_ = (σ^2^P_1_ + σ^2^P_2_)/2, σ^2^_p_ = F_2_ variance, σ^2^_g_ = σ^2^_p_ − σ^2^_E_, and k is the selection intensity from selecting 10% of the superior plants.

Heritability in the narrow sense (h^2^) was calculated as in [29], and estimated with parent–offspring regression.

Significance of the observed direct and correlated genetic advance with selection in percentage from the mid- and the better parent, was measured using a least significant differences (LSD) test, to clarify if the selection produced an actual genetic advance or not.
Observed genetic advance from the better parent = (population mean − better parent)/better parent) × 100
Observed genetic advance from the mid parent = (population mean − mid parent)/mid parent) × 100

## 5. Conclusions

Selection mainly depends upon the presence of genetic variability in the traits concerned. After two cycles of selection, the genetic variability in the selection criteria decreased. However, the remaining genetic variability in the selection criteria was sufficient for further cycles of selection for HFC, LFZ, and SYP^−1^. Inclusion of SYP^−1^ with 50% flowering, PH, and LFZ as a restricted trait preserved genetic variability better than single trait selection. Unreliable estimates of heritability in the broad sense were obtained in the F_4_-generation for the selection criteria, which ranged from 87.92% to 97.80%, because of the evaluation of the selected families at one site for one-year inflated families mean squares by the cofounding effect of the interactions of families with locations and years. However, the narrow-sense heritability (h^2^), as estimated by parent–offspring regression of F_4_/F_3_, was low to moderate. Single trait selection was effective for improving the selection criterion, but it caused an unfavorable decrease in some correlated traits. The highest observed genetic gain in SYP^−1^ was obtained from selection for LFZR, followed by selection for PHR, SYP^−1^, PH, LFZ, and FLOWR. The correlated observed gain in infection% was significant (*p* ≤0.01) from the mid-parent, and ranged from −20.77 to −27.32% in pop1, and from −7.02 to −11.58% in pop2. Therefore, a selection index which involves yield and its components can be recommended to solve this problem.

## Figures and Tables

**Figure 1 plants-11-01538-f001:**
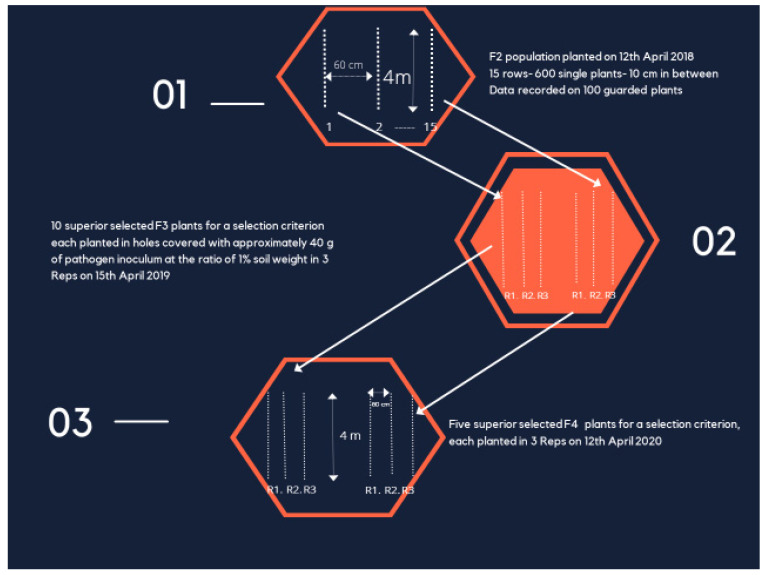
Demonstration of the two cycles of selection in F_2_, F_3_, and F_4_- generations.

**Table 1 plants-11-01538-t001:** Statistical analysis summary of the populations, including means of the studied traits, genotypic (GCV) and phenotypic (PCV) coefficients of variation, heritability in the broad sense (Hb%), and expected genetic advance (ΔG) of the two bases populations in the F_2_- generation evaluated under artificial infection of *F. oxysporum*.

Item	50% Flow (Days)	PH (cm)	HFC (cm)	LFZ (cm)	SYP^−1^ (g)	Inf (%)
Population	*Shandaweel 3/Sohag2000* (pop1)					
Mean ± SE	65.80 ± 0.55	130.50 ± 1.78	55.80 ± 1.44	74.70 ± 1.54	10.50 ± 0.62	60.32
Max.	85.00	175.00	105.00	115.50	27.60	
Min.	48.40	88.50	28.10	33.60	4.70	
GCV%	4.6	10.81	22.83	15.04	33.60	
PCV%	8.42	13.61	25.90	20.56	59.17	
Hb%	29.89	63.05	77.72	53.54	32.25	
ΔG	2.91	19.72	19.77	14.47	3.53	
ΔG/mean%	4.43	15.11	35.42	19.37	33.59	
Parent	*Shandaweel3*					
Mean ± SE	59.6 ± 1.50	118.5 ± 3.50	60.5 ± 2.32	58 ± 3.98	11.8 ± 1.60	33.57
Max.	63.20	135.10	70.40	77.10	15.50	
Min.	49.90	98.90	53.40	35.60	4.60	
CV%	7.96	9.35	12.13	21.71	42.88	
Parent	*Sohag2000*					
Mean ± SE	55.4 ± 1.43	90.2 ± 3.32	37.1 ± 1.98	53.1 ± 2.46	9.7 ± 1.63	76.5
Max.	64.50	100.20	45.60	70.10	12.90	
Min.	50.50	72.40	21.40	38.50	6.20	
CV%	8.18	11.65	16.87	14.65	53.27	
Population	*Int.562/Int.688* (pop2)					
Mean ± SE	57.25 ± 0.83	114.48 ± 2.23	42.08 ± 1.36	72.4 ± 1.89	10.62 ± 0.77	63.7
Max.	68.00	170.00	80.00	120.00	42.56	
Min.	40.00	60.00	10.00	23.00	4.70	
GCV%	12.58	16.68	10.35	20.54	58.84	
PCV%	14.50	19.44	32.21	26.05	72.17	
Hb%	75.20	73.65	10.32	62.14	66.52	
ΔG	10.99	28.85	2.46	20.63	8.97	
ΔG/mean%	19.20	25.20	5.85	28.49	84.50	
Parent	*Int.562*					
Mean ± SE	68.5 ± 1.28	107 ± 3.88	50 ± 3.59	57.6 ± 3.18	10.6 ± 1.37	65.71
Max.	70.00	135.00	63.00	75.00	14.75	
Min.	56.00	83.00	35.00	40.00	6.70	
CV%	5.93	11.46	22.69	17.43	40.90	
Parent	*Int.688*					
Mean ± SE	58.7 ± 1.33	110.75 ± 3.32	40 ± 4.48	70.75 ± 4.10	12.45 ± 1.43	55.45
Max.	59.00	135.00	60.00	85.00	14.60	
Min.	53.00	95.00	20.00	50.00	7.80	
CV%	7.17	9.49	35.42	18.35	36.37	

ΔG = expected genetic advance from selection 10% superior plants, PH = plant height, HCF = height to first capsule, LFZ = length of the fruiting zone, SYP^−1^ = seed yield plant^−1^, Inf% = infection%.

**Table 2 plants-11-01538-t002:** Phenotypic correlation coefficients among traits in the F_2_- generation, *Sohag2000/Shand3* (pop1) above and *Int526/int688* (pop2) population below diagonal.

	50% Flow (Days) Flow	PH	HFC	LFZ	SYP^−1^	1000SW
**50% Flow**		−0.33 **	−0.01	−0.40 **	−0.56 **	−0.04
**PH**	0.10		0.61 **	0.60 **	0.41 **	−0.34 **
**HFC**	0.23 *	0.52 **		−0.27 **	0.04	−0.21 *
**LFZ**	−0.04	0.78 **	−0.13		0.46 **	−0.20 *
**SYP^−1^**	−0.27 *	0.78 **	0.02	0.50 **		−0.01
**1000SW**	−0.05	−0.40 **	−0.30 **	−0.23 *	−0.01	

PH = plant height, HFC = height to 1st capsule, LFZ = length of the fruiting zone, SYP^−1^ = seed yield plant^−1^, *, ** significant at 0.05 and 0.01 levels of probability, respectively.

**Table 3 plants-11-01538-t003:** Pertinent mean squares of the studied traits for the two populations in the F_4_- generation.

Sel. C. §	Population	50% Flow	PH	HFC	LFZ	NCP^−1^	SYP^−1^	1000SW	Inf%
50% Flow	*Sohag2000* */Shand3*	11.87 **	48.41 *	117.32 **	53.27	641.99 **	2.81 **	0.3 **	293.16 **
		0.45	15.20	13.51	59.34	21.64	0.45	0.02	1.75
	*Int.512* */Int. 688*	33.75 **	513.30 **	362.10 **	224.41 *	892.45 **	23.59 **	0.26 **	55.34 **
		3.06	53.23	21.69	59.26	31.41	0.52	0.02	0.67
PH	*Sohag2000* */Shand3*	318.86 **	688.49 **	96.21 **	785.65 **	1310.44 **	13.35 **	0.57 **	24.21 **
		7.21	18.45	9.56	29.48	22.85	0.80	0.017	0.67
	*Int.512* */Int. 688*	175.56 **	256.43 **	46.76	396.78 **	232.20 **	6.63 **	0.083 **	128.2 **
		3.90	30.98	21.38	56.64	27.56	0.59	0.02	2.54
HFC	*Sohag2000* */Shand3*	41.73 **	547.09 **	247.19 **	217.52 **	815.29 **	23.59 **	0.50 **	5.99 **
		4.05	41.32	10.10	20.91	24.34	1.50	0.02	1.26
	*Int.512* */Int. 688*	31.86 **	239.38 **	91.72 **	102.99	1248.13 **	27.63 **	0.39 **	84.90 **
		2.477	18.92	8.12	41.88	42.12	0.35	0.02	0.78
LFZ	*Sohag2000* */Shand3*	341.05 **	882.54 **	371.19 **	624.05 **	1093.93 **	55.41 **	0.47 **	66.33 **
		10.83	23.60	24.41	33.33	29.78	0.85	0.01	0.81
	*Int.512* */Int. 688*	114.60 **	327.19 **	45.52	300.68 **	325.98 **	6.05 *	0.05 *	79.72 **
		6.82	20.31	23.83	38.56	36.63	0.71	0.01	2.30
SYP^−1^	*Sohag2000* */Shand3*	29.30 **	233.33 **	368.97 **	1280.56 **	1333.10 **	36.85 **	0.11 **	68.27 **
		1.65	20.2	21.873	63.936	18.1	0.81	0.013	1.89
	*Int.512* */Int. 688*	73.49 **	676.45 **	75.05	631.38 **	1264.20 **	7.71 **	0.035	89.20 **
		1.5	34.78	26.67	26.1	28.31	0.321	0.019	1.69
FlOW (R)	*Sohag2000* */Shand3*	253.87 **	929.32 **	397.71 **	883.75 **	568.21 **	13.16 **	0.11 **	193.16 **
		8.087	24.32	20.95	59.341	28.9	2.64	0.01	2.55
	*Int.512* */Int. 688*	52.21 **	370.44 **	72.44 *	252.67 **	213.92 **	5.62 **	0.03	58.38 **
		1.44	32.25	21.73	48.33	54.99	0.48	0.022	0.68
PHR	*Sohag2000* */Shand3*	320.60 **	1887.43 **	96.27 **	1949.32 **	2657.56 **	28.55 **	0.47 **	28.23 **
		3.46	23.75	14.56	39.66	40.82	3.424	0.0085	0.68
	*Int.512* */Int. 688*	116.82 **	235.52 **	55.52	388.76 **	991.39 **	6.88 **	0.07 *	130.5 **
		7.730159	29.71429	19.9	57.78	37.85	0.61	0.02	3.60
LFZR	*Sohag2000* */Shand3*	41.63 **	1107.41 **	96.27 **	1710.21 **	2633.49 **	3.22 **	0.12 **	68.43 **
		2.78	27.89	18.6	38.42	35.78	0.852	0.0121	0.83
	*Int.512* */Int. 688*	76.11 **	654.85 **	76.19 *	603.76 **	1008.05 **	6.11 **	0.03	75.73 **
		7.25	31.02	21.17	42.78	39.19	0.75	0.02	2.31

* and **; significant at 0.05 and 0.01 level of probability, respectively, Sel. C. § = selection criterion, PH = plant height, HCF = height of the first capsule, LFZ = length of the fruiting zone, NCP^−1^ = number of capsules plant^−1^, SYP^−1^ = seed yield plant^−1^, 1000SW = 1000 seed weight, 50% (FLOWR) = days to 50% flow restricted by seed yield, PHR = plant height restricted by seed yield, LFZR = LFZ restricted by seed yield, Inf% = infection%.

**Table 4 plants-11-01538-t004:** Genotypic (GCV%) and phenotypic (PCV%) coefficients of variation for the studied traits for the two populations in the F_4_- generation.

Sel. C.	Population	Item	50% Flow	PH (cm)	HFC (cm)	LFZ (cm)	NCP^−1^	SYP^−1^ (g)	1000SW (g)	Inf (%)
50% Flow	*Sohag2000* */Shand3*	GCV%	3.68	3.20	13.27	ns	24.21	9.16	9.52	27.73
		PCV%	3.75	3.86	14.11	ns	24.63	10.00	9.76	27.81
	*Int.512* */Int. 688*	GCV%	5.70	12.88	24.47	14.11	26.33	30.03	7.50	9.33
		PCV%	5.98	13.61	25.24	16.44	26.81	30.37	7.82	9.38
PH	*Sohag2000* */Shand3*	GCV%	15.73	11.47	12.14	20.00	19.71	16.27	11.38	7.44
		PCV%	15.91	11.62	12.79	20.38	19.88	16.78	11.55	7.55
	*Int.512* */Int. 688*	GCV%	11.63	7.28	6.68	14.08	8.48	12.36	3.62	14.24
		PCV%	11.76	7.76	9.07	15.21	9.03	12.95	4.15	14.39
HFC	*Sohag2000* */Shand3*	GCV%	6.46	13.80	19.02	17.10	29.75	32.54	11.59	3.62
		PCV%	6.80	14.36	19.42	17.99	30.20	33.62	11.83	4.08
	*Int.512* */Int. 688*	GCV%	5.45	8.77	12.35	ns	25.32	35.28	10.03	11.38
		PCV%	5.67	9.13	12.94	ns	25.75	35.51	10.30	11.44
LFZ	*Sohag2000* */Shand3*	GCV%	15.80	14.18	22.56	19.58	19.00	34.66	9.96	13.41
		PCV%	16.06	14.37	23.34	20.12	19.26	34.93	10.03	13.49
	*Int.512* */Int. 688*	GCV%	9.36	8.62	ns	12.19	9.39	11.41	2.54	11.48
		PCV%	9.65	8.90	ns	13.66	9.97	12.15	3.05	11.65
SYP^-1^	*Sohag2000* */Shand3*	GCV%	5.23	7.16	22.89	31.80	21.38	26.99	4.39	13.23
		PCV%	5.38	7.50	23.60	32.62	21.52	27.30	4.66	13.42
	*Int.512* */Int. 688*	GCV%	7.74	12.81	ns	19.91	19.74	13.30	ns	12.00
		PCV%	7.82	13.15	ns	20.34	19.97	13.58	ns	12.12
FLOWR	*Sohag2000* */Shand3*	GCV%	14.97	15.54	21.33	11.94	18.57	18.62	4.39	25.75
		PCV%	15.21	15.74	21.92	14.10	19.06	20.82	4.71	24.85
	*Int.512* */Int. 688*	GCV%	7.09	10.24	10.05	13.15	8.05	11.71	ns	9.35
		PCV%	7.19	10.72	12.01	14.62	9.34	12.24	ns	10.38
PHR	*Sohag2000* */Shand3*	GCV%	16.82	21.39	11.82	34.85	32.38	23.72	9.96	8.50
		PCV%	16.91	21.52	12.8	35.21	32.63	25.29	10.06	7.00
	*Int.512* */Int. 688*	GCV%	8.52	6.54	ns	12.76	16.51	11.77	3.30	13.20
		PCV%	8.82	7.00	ns	13.83	16.83	12.33	3.87	12.40
LFZR	*Sohag2000* */Shand3*	GCV%	6.38	16.10	11.53	35.59	29.54	6.27	4.63	14.50
		PCV%	6.60	16.31	12.84	35.99	29.75	7.31	4.87	12.60
	*Int.512* */Int. 688*	GCV%	7.16	12.14	9.87	18.15	16.98	11.10	ns	10.45
		PCV%	7.52	12.44	11.61	18.83	17.31	11.85	ns	10.60

ns; insignificant differences among families, Sel. C. = selection criterion. PH = plant height, HCF = height to the first capsule, LFZ = length of fruiting zone, NCP^−1^ = number of capsules plant^−1^, SYP^−1^ = seed yield plant^−1^, 1000SW = 1000 seed weight, 50% FLOWR = 50% flow restricted by seed yield, PHR = plant height restricted by seed yield, LFZR = LFZ restricted by seed yield, Inf% = infection%, ns = insignificant differences among families, restricted = (the best family in the concerned trait which gave seed yield plant^−1^ equal to or more than the population mean).

**Table 5 plants-11-01538-t005:** Heritability in the broad (H%) and narrow sense (h^2^) as estimated by parent–offspring regression (F_4_/F_3_) for the selection criteria for both populations.

Population	Shan.3/Sohag2000		Int.562/Int.688	
Sel. criterion	H%	h^2^	H%	h^2^
50% Flow.	96.21	0.48	90.94	0.49
Plant height	97.32	0.40	87.92	0.37
HFC	95.91	0.44	91.15	0.42
LFZ	94.66	0.22	79.64	0.22
SYP^−1^	97.80	0.29	95.84	0.22
FLOWR	96.82	0.32	97.23	0.34
PHR	98.92	0.27	93.38	0.28
LFZR	93.33	0.22	90.47	0.24

HCF = height to the first capsule, LFZ = length of fruiting zone, SYP^−1^ = seed yield plant^−1^, FLOWR = 50% flow restricted by SYP^−1^, PHR = plant height restricted by SYP^−1^, LFZR = LFZ, restricted by SYP^−1^, restricted = (the best family in the concerned trait, which gave a seed yield plant^−1^ equal to or more than the population mean).

**Table 6 plants-11-01538-t006:** The observed genetic advance (GA) in percentage from the mid-parent (GA. MP%) and better parents (GA. BP%) after two cycles of pedigree selection for the two populations.

Population	Sel. C.	Item	50% Flow	PH (cm)	HFC (cm)	LFZ (cm)	NCP^−1^	SYP^−1^ (g)	1000SW (g)	Inf (%)
*Sohag2000/* *Shand.3*	50% Flow	Mean	53.07	104.00	44.33	59.67	59.40	9.67	3.59	35.55
		GA.MP%	−4.96 **	−4.00	−14.74 *	ns	−31.85 **	−6.04	−9.03 **	−25.16 **
		GA.BP%	−2.33 *	−6.87 *	−9.52	ns	−36.36 **	−15.6 **	−14.44 **	−1.72
*INT.562/* *INT.688*		Mean	56.07	96.13	43.53	52.60	64.33	9.23	3.77	45.78
		GA.MP%	−8.59 **	−12.61 *	−11.76	−13.30	−27.96 **	−8.12	−3.41	−8.49 **
		GA.BP%	−6.56 *	−11.80 *	−7.38	−15.16	−30.91 **	−13.88 *	−4.87	−2.10
*Sohag2000/* *Shand.3*	PH	Mean	64.80	130.33	44.27	79.40	105.13	12.57	3.78	37.63
		GA.MP%	16.06 **	20.31 **	−14.87 **	40.95 **	20.61 **	22.13 **	−4.30	−20.77 **
		GA.BP%	19.26 **	16.72 **	−9.66	40.12 **	12.64 *	9.65	−10.00 **	4.04 *
*INT.562/* *INT.688*		Mean	65.07	119.13	43.53	75.60	97.40	11.48	4.01	45.44
		GA.MP%	6.09 *	8.30 *	−11.76	24.62 *	9.07 *	14.23 *	2.56	−9.17 **
		GA.BP%	8.44 **	9.30 *	−7.38	21.94 *	4.61	7.07	1.01	−2.82
*Sohag2000/* *Shand.3*	HFC	Mean	54.87	94.07	46.73	47.33	54.58	8.34	3.45	34.68
		GA.MP	−1.73	−13.17 **	−10.13 *	−15.98 *	−37.39 **	−18.99 *	−12.74 **	−26.98 **
		GA.BP	0.98	−15.76 **	−4.63	−16.47 *	−41.52 **	−27.27 **	−17.94 **	−4.11
*INT.562/* *INT.688*		Mean	57.47	97.80	42.73	55.07	79.20	8.55	3.52	46.52
		GA.MP%	−6.30 **	−11.09 **	−13.38 **	ns	−11.31 *	−14.96 **	−9.90 **	−7.02 **
		GA.BP%	−4.22	−10.28 **	−9.08	ns	−14.94 *	−20.29 **	−11.26 **	−0.52
*Sohag2000/* *Shand.3*	LFZ	Mean	66.40	119.33	47.67	71.67	99.13	12.30	3.93	34.86
		GA.MP%	18.93 **	10.15 **	−8.33	27.22 **	13.73 **	19.51 **	−0.42	−26.61 **
		GA.BP%	22.21 **	6.87	−2.72	26.47 **	6.21	7.30	−6.35 **	−3.63
*INT.562/* *INT.688*		Mean	64.07	117.33	44.07	73.27	104.60	11.69	4.06	44.23
		GA.MP%	4.46	6.67 *	ns	20.77 *	17.13 **	16.28 *	3.92	−11.58 **
		GA.BP%	6.78	7.65 *	ns	18.17 *	12.34 *	8.99	2.35	−5.40
*Sohag2000/* *Shand.3*	SYP^−1^	Mean	58.07	117.67	47.00	63.33	97.93	12.84	4.18	35.55
		GA.MP%	4.00 *	8.62 *	−9.62	12.43	14.54 **	24.72 **	5.82 *	−25.16 **
		GA.BP%	6.87 **	5.37	−4.08	11.76	8.81 *	11.98 *	−0.48	−1.72
*INT.562/* *INT.688*		Mean	63.27	114.20	42.87	71.33	102.80	11.80	4.05	45.01
		GA.MP%	3.15 *	3.82	ns	17.58 *	15.12 **	17.43 **	ns	−10.04 **
		GA.BP%	5.44 **	4.77	ns	15.05 *	10.41 *	10.07 *	ns	−3.75
*Sohag2000/* *Shand.3*	50% FLOWR	Mean	54.0	118.0	52.53	70.00	95.0	12.10	4.07	34.86
		GA.MP%	−3.28	8.92 *	1.02	24.26 *	11.11 *	17.53 *	3.12	−26.61 **
		GA.BP%	−0.62	5.67	7.21	23.53	5.56	5.52	−3.02	−3.63
*INT.562/* *INT.688*		Mean	58.0	118.60	40.92	71.80	105.60	11.20	3.98	45.00
		GA.MP%	−5.43 *	7.82 *	−17.06 *	18.35 *	18.25 **	11.44 *	1.96	−10.04 **
		GA.BP%	−3.33	8.81 *	−12.94	15.81	13.41	4.40	0.42	−3.75
*Sohag2000/* *Shand.3*	PHR	Mean	61.13	130.50	44.13	80.50	105.00	13.5	3.93	36.73
		GA.MP%	9.49 **	20.46 **	−15.12 *	42.90 **	22.81 **	31.12 *	−0.59	−22.67 **
		GA.BP%	12.52 **	16.86 *	−9.93	42.06 **	16.67 *	17.73	−6.51 **	1.54
*INT.562/* *INT.688*		Mean	70.73	120.50	44.33	80.50	108.00	12.29	3.93	49.74
		GA.MP%	15.33 **	9.54 *	−10.14	32.69 **	20.94 **	22.25 **	0.68	−10.04 **
		GA.BP%	17.89 **	10.55 *	−5.67	29.84 *	15.99 *	14.59 *	−0.84	6.38
*Sohag2000/* *Shand.3*	LFZR	Mean	56.40	125.30	44.13	73.7	105.00	13.80	4.18	34.52
		GA.MP%	1.01 **	15.66 **	−15.13 *	30.83 **	22.81 **	34.04 **	5.82 *	−27.32 **
		GA.BP%	3.83 **	12.21 *	−9.93	30.06 **	16.67 **	30.67 **	−0.48	−4.55
*INT.562/* *INT.688*		Mean	66.93	120.5	43.4	75.80	105.87	12.90	4.01	47.59
		GA.MP%	9.13 *	9.54 *	−12.03	24.94 **	18.55 **	28.35 **	2.73	−10.04 **
		GA.BP%	11.56 **	10.55 *	−7.66	22.26 *	13.70 *	20.31 **	1.18	1.77

* and **; significant at a 0.05 and 0.01 level of probability, respectively, ns; insignificant differences among entries, Sel. C. = selection criterion, PH = plant height, HCF = height to first capsule, LFZ = length of fruiting zone, NCP^−1^ = number of capsules plant^−1^, SYP^−1^ = seed yield plant^−1^, 1000SW = 1000 seed weight, FLOWR = flowering restricted by SYP^−1^, PHR = plant height restricted by yield, LFZR = length of fruiting zone restricted by yield, Inf% = infection %.

## Data Availability

Not applicable.

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
