# Peer review of "Improving Sesame (Sesamum indicum L.) Seed Yield through Selection under Infection of Fusarium oxysporum f. sp. sesami"

_plants, 2022, doi:10.3390/plants11121538_

Round 1

Reviewer 1 Report

In general the manuscript was well prepared. However minor errors need to be

corrected. Firstly ALL scientific names must be in italics. The authors names should be added to F. sesame f.sp. seasame when first used but not needed later on. Secondly it is not appropriate to start a sentence with the reference citation number. Thus the authors must be inserted before the number as on

lines 35, 44, 46, 53, 55, 272, 273,307, 331, 343, 344, 355, 356.

line 54 add and before "exceeded"

line 57 pidigree should be  pedigree

line 58 add "ly" to "sigificant"

line 59 should be "plant"

line 118, 152 add space between pop1 and indicating; between SYP1 and as

line 189, 302 didn't should be spelled out as did not

line 255 add "the previous findings" before the citations.

line 262, 320 add "and" after "heteroxygosity"; add "and" after the citations

line 264, 285, 320 delete "by" before the citations.

line 386 change , to ;

line 387, add "was" before "imported"

line 424 add "reported" after "as"

line 429 replace date was a citation number'

line 430 delete "Where"

line add "reported"  after "as

line 434 add authors before the citation

line 439 "didn't" should be "did not"

Author Response

Dear Editor,

Thank you for editing. I am Pleased and honored with your findings in the manuscript and hope you find the new attachment well written and satisfy the best form of the paper.

A new diagram is inserted and also new paragraphs for better manuscript data description (This was demanded by the second editor )

All scientific names were edited, for the first point.

about the author names cited in the manuscript: in the journal style the author name cited is referred to as a number. I changed the position of the citations as much as possible in the manuscript.

thank you so much for your editing and hope this time the manuscript satisfies you.

Reviewer 2 Report

Abstract: Rewrite it to improve understanding. Add the aims.

Keywords: Change “Sesamum indicun L, Fusarium oxysporum” because they are in the title.

Introduction:

It is poor. Add more research about this study or the pathogen, or the plant, etc.

Lines 46, 50, 53. 55: Who have evaluated or found, etc? Add a subject to the sentences.

Results:

Lines 68-71: What does it mean “PH, LFZ, HFC or SYP? Because they are acronyms that are the first time they appear in the text. Write the complete name and next to it the acronym.

Tables should be placed in the main text near to the first time they are cited, for example Table 6.

Table 1: Correct the footer.

Line 118: Is “FZ” or “LFZ”?

Discussion

Rewrite it. There are few studies that compare the data. They do not discuss the data but repeat the results without clarifying them. It's hard to follow.

Conclusions: They should be after material and methods section.

Material and methods:

I suggest adding a graphic to explain the process.

Complete this section, it is very poor. Add how it was inoculated the pathogen.

Was it used the oil percentage in the research? I have not seen those results.

Lines 421, 424, 432: Who have evaluated or found, etc? Add a subject to the sentences

The authority for a Latin name should be provided after the first time it is referred to in the title, abstract, main body, and a figure or table description.

References and ALL text: scientific names are cited in italic; these names must be different of the format of the section. Adapt the references to journal standards. The genus, species, and variety name, genes must write in italics. Check all text.

Author Response

Dear Editor,

 Honored and pleased with your editing thank you so much for such detailed review, I hope this time you find the manuscript well written and deserve publishing after your mentioned points.

Abstract: The aims were added and it was rewritten

Keywords: done

introduction: rewritten and more researches added in blue 

Results: all done

Table 6: relocated

Table 1: footer is corrected

Line 118: it is LFZ and corrected

Discussion: rewritten 

Conclusion: relocated

Material and methods: a diagram was added, hope you find it well describing.

This section is completed and a whole paragraph about inoculating the pathogen is added as you kindly pointed to.

The oil percentage was non significant and thus deleted, this was also added in the text.

All the lain names were corrected.

References were corrected.

Hope this time you find it well, and thank you for helping in improving this manuscript.

Round 2

Reviewer 2 Report

Fusarium is a genus, and it must write in italics. Check all text, including references.

Line 48: F. oxysporum f.sp. sesami, => sesami has to write in italics.

Author Response

Dear Reviewer,

Glad and Honored with your editing, and hope this manuscript full fill your satisfaction.

All changes have been done and wish you the best as we have learned from you a lot. 
